# Bioactive Properties of Microencapsulated Anthocyanins from *Vaccinium floribundum* and *Rubus glaucus*

**DOI:** 10.3390/molecules29235504

**Published:** 2024-11-21

**Authors:** Carlos Barba-Ostria, Rebeca Gonzalez-Pastor, Fabián Castillo-Solís, Saskya E. Carrera-Pacheco, Orestes Lopez, Johana Zúñiga-Miranda, Alexis Debut, Linda P. Guamán

**Affiliations:** 1Instituto de Microbiología, Colegio de Ciencias Biológicas y Ambientales COCIBA, Universidad San Francisco de Quito, Quito 170901, Ecuador; cbarbao@usfq.edu.ec; 2Escuela de Medicina, Colegio de Ciencias de la Salud, Universidad San Francisco de Quito, Quito 170901, Ecuador; 3Centro de Investigación Biomédica, Facultad de Ciencias de la Salud Eugenio Espejo, Universidad UTE, Quito 170527, Ecuador; rebeca.gonzalez@ute.edu.ec (R.G.-P.); saskyacarrera@gmail.com (S.E.C.-P.); johana.zuniga@ute.edu.ec (J.Z.-M.); 4Facultad de Ciencia e Ingeniería en Alimentos, Universidad Técnica de Ambato, Ambato 180206, Ecuador; fabian227@outlook.com (F.C.-S.); od.lopez@uta.edu.ec (O.L.); 5Centro de Nanociencia y Nanotecnología, Universidad de Las Fuerzas Armadas ESPE, Sangolquí 171103, Ecuador; apdebut@espe.edu.ec; 6Departamento de Ciencias de la Vida y Agricultura, Universidad de las Fuerzas Armadas ESPE, Sangolquí 171103, Ecuador

**Keywords:** anthocyanins, microencapsulation, antioxidant activity, phenolic compounds, antitumoral activity, antibacterial, SDG 3, good health and well-being

## Abstract

Anthocyanins, widely recognized for their antioxidant properties and potential health benefits, are highly susceptible to degradation due to environmental factors such as light, temperature, and pH leading to reduced bioavailability and efficacy. Microencapsulation, which involves entrapment in a matrix to enhance stability and bioavailability. This study aims to investigate the bioactive properties of microencapsulated anthocyanins derived from *Vaccinium floribundum* (Andean blueberry) and *Rubus glaucus* (Andean blackberry). The extracts from *V. floribundum* and *R. glaucus* were microencapsulated using maltodextrin as the carrier agent due to its film-forming properties and effectiveness in stabilizing sensitive compounds through a spray-drying process. The microcapsules were characterized using Fourier Transform Infrared Spectroscopy (FTIR) and Scanning Electron Microscopy (SEM) to assess their chemical and morphological properties. The biological activities of these microencapsulated anthocyanins were evaluated using in vitro assays for their antibacterial, antioxidant, and anti-inflammatory effects. The results indicated enhanced bioactivity of the microencapsulated anthocyanins, suggesting their potential use in developing functional foods and pharmaceuticals. This study provides valuable insights into the effectiveness of microencapsulation in preserving anthocyanins’ functional properties and enhancing their health-promoting effects, highlighting the potential for application in the food and pharmaceutical industries.

## 1. Introduction

Anthocyanins are widely present in fruits and vegetables such as blueberries, blackberries, and red cabbage, contributing to their vibrant colors and health benefits [1]. These naturally occurring pigments have drawn considerable attention in recent years due to their powerful antioxidant properties and potential health benefits. Specifically, anthocyanins have been associated with the prevention of various chronic diseases, including cancer, cardiovascular diseases, and neurodegenerative disorders [2,3,4]. Despite their beneficial properties, anthocyanins are highly susceptible to degradation when exposed to environmental factors such as light, temperature, and pH, which can significantly affect their stability and bioavailability [5,6].

To address these challenges, microencapsulation has emerged as a promising technique for protecting anthocyanins from environmental degradation and to control their release in the body [7]. Microencapsulation not only protects anthocyanins from degradation but also controls their release in the body, allowing for a sustained release and absorption over time [8]. In this study, we focus on the microencapsulation of anthocyanins extracted from *Vaccinium floribundum* (Andean blueberry) and *Rubus glaucus* (Andean blackberry), two fruits native to Ecuador that are known for their high anthocyanin content and potential health benefits [9]. Examining their encapsulation and bioactivity in complex matrices introduces a novel perspective, adding valuable insights into local sources of anthocyanins with rich bioactive properties.

The primary objectives of this research were to extract and characterize anthocyanins from *V. floribundum* and *R. glaucus*, microencapsulate the extracts using maltodextrin as a carrier agent, and to evaluate the biological activities of the microencapsulated products. Specifically, we aimed to assess the antibacterial, antioxidant, and cytotoxic effects of the microencapsulated anthocyanins to determine their potential applications in the development of functional foods and pharmaceutical development.

The microencapsulation process was carried out using a spray-drying method with maltodextrin as the encapsulating agent. The resulting microcapsules were characterized using FTIR and SEM to determine their chemical and morphological properties. The biological activities of the microencapsulated anthocyanins were assessed through a series of assays evaluating their antibacterial, antioxidant, and anti-inflammatory effects.

Recent studies on anthocyanin microencapsulation have focused on addressing challenges related to stability, bioavailability, and controlled release, but limitations in encapsulation efficiency and the influence of different encapsulating agents persist [10]. This study seeks to improve these aspects by exploring the use of maltodextrin as an encapsulating matrix in the spray-drying process. Maltodextrin offers significant advantages, including its ability to form a stable barrier that protects anthocyanins from environmental stressors [11]. Additionally, the spray-drying process was chosen for its scalability and capacity to produce uniform microspheres, making it suitable for industrial applications.

By demonstrating the enhanced bioactivity of microencapsulated anthocyanins, this research highlights their applicability in the development of novel pharmaceutical and nutraceutical products.

## 2. Results and Discussion

### 2.1. Chemical Characterization and Antioxidant Activity

The phytochemical composition and antioxidant potential of *V. floribundum* and *R. glaucus* fruits were assessed by quantifying their total polyphenol content (TPC), anthocyanin levels, and antioxidant activity via the DPPH assay.

The comparative analysis of *V. floribundum* and *R. glaucus* in Table 1 reveals significant differences in their bioactive compounds and antioxidant capacities. Specifically, *V. floribundum* exhibited a total polyphenol content of 354 ± 25.16 mg/g fresh weight, substantially higher than *R. glaucus*’s 294 ± 24.03 mg/g. Similarly, the anthocyanin content in *V. floribundum* was 79.67 mg/100 g, surpassing *R. glaucus*’s 53.3 mg/100 g. These differences are reflected in their antioxidant activities, with *V. floribundum* demonstrating a stronger capacity (83.5 ± 19.40 μg/mL) compared to *R. glaucus* (167.92 ± 39.57 μg/mL).

In comparison with the existing literature, studies have reported high polyphenol and anthocyanin contents in *V. floribundum*, which supports its classification as a “superfruit” due to its potent antioxidant properties [12]. For instance, similar studies on *V. floribundum* have found total polyphenol contents ranging from 150 to 300 mg/100 g and anthocyanin levels between 35 and 300 mg/100 g, aligning closely with the present data [13]. These compounds contribute significantly to its high antioxidant activity, which is essential for its health-promoting properties.

Although the previous literature has reported polyphenol contents of approximately 400 mg/100 g for some *Rubus* species, we found that the polyphenol content was 294 ± 24.03 mg/100 g. This indicates a lower value, which is also reflected in the low anthocyanin content (53.3 mg/100 g) [14,15]. Although the fruits used in this study were purchased from a market, it is possible that the supply chain conditions, such as time and temperature between harvest and display, may have influenced the polyphenol and anthocyanin content. However, the observed differences align with variations previously reported in the literature, which can be attributed to factors like cultivation conditions and maturation [16,17]. Future studies could further refine these results by using freshly harvested fruits to ensure optimal retention of bioactive compounds.

Given their high concentration of polyphenols and anthocyanins, berries hold potential for pharmacological applications, in the treatment of diseases associated with oxidative stress and inflammation, which have demonstrated potential cancer chemopreventive activity, as confirmed by human clinical trials [18,19].

In terms of antioxidant potential, both berries showed up to 43.7-fold less DPPH scavenging activity than the ascorbic acid control (IC_50_ 3.84 ± 0.92 μg/mL); however, *V. floribundum* exhibited a 2-fold higher anthocyanin content than *R. glaucus*. The higher polyphenol and anthocyanin contents in *V. floribundum* correlate with the enhanced antioxidant activity determined by the DPPH assay. The superior antioxidant activity of *V. floribundum* was also observed in our previous study of non-microencapsulated berries [9]; however, in that study, the IC_50_ values for the two fruits were very similar and at least 3.8-fold lower than in the current study. The *t*-test results showed significant differences between *V. floribundum* and *R. glaucus* for TPC, anthocyanin content, and antioxidant activity (*p* < 0.001 in all cases). This suggests that the microencapsulation process or the removal of the maltodextrin prior to the DPPH scavenging determination affected the overall antioxidant potential [20,21]. Interestingly, the literature often highlights the benefits of microencapsulation to preserve the antioxidant activity of natural extracts [22,23,24]. Therefore, additional antioxidant studies using other methods that are compatible with the maltodextrin matrix should be explored to confirm these findings.

### 2.2. Fourier Transform Infrared Spectroscopy (FTIR) Analysis

The FTIR spectra analysis of non-microencapsulated and microencapsulated anthocyanins extracted from *Vaccinium floribundum* (Andean blueberry) and *Rubus glaucus* (Andean blackberry) provides significant insights into the structural and functional properties of these compounds. The presence of characteristic peaks indicative of various functional groups offers a comprehensive understanding of the anthocyanins’ chemical composition and the impact of microencapsulation, as summarized in Table 2 and Table 3, and illustrated in Appendix A.

The analysis confirmed the successful microencapsulation of anthocyanins from both berry species, with the presence and modification of key functional group peaks in the FTIR spectra verifying the encapsulation process. As shown in Appendix A and Table 2 and Table 3, the FTIR spectra of anthocyanins in panel A depict the non-microencapsulated extracts, while panel B represents the microencapsulated microspheres for both *V. floribundum* and *R. glaucus*. Notably, distinct bands were observed around 1637 cm^−1^ and 1635 cm^−1^ in the extracts, corresponding to the O-H stretching of polyphenols. In the microencapsulated anthocyanins, these bands were significantly reduced or absent, indicating an efficient encapsulation process that likely reduces the exposure of reactive groups, thus enhancing stability against environmental stressors such as pH, temperature, and light exposure [25].

The encapsulation process likely plays a key role in reducing anthocyanin degradation by limiting the exposure of reactive hydroxyl groups, which are highly susceptible to environmental stressors such as light, pH, and temperature [26]. The FTIR spectra revealed a significant reduction or absence of O-H stretching bands in the microencapsulated anthocyanins, indicating that the matrix (maltodextrin) may act not only as a physical barrier but also potentially engage in hydrogen bonding with anthocyanins, further stabilizing them [11]. This interaction between the matrix and the anthocyanins could contribute to reducing reactivity and enhancing stability. While our FTIR analysis suggested a decrease in the exposure of the reactive hydroxyl groups, likely due to the formation of hydrogen bonds, no major chemical modifications to the core anthocyanin structure were detected. Maltodextrin primarily serves as a protective barrier against environmental factors, stabilizing anthocyanins and slightly altering their local structure without affecting their overall functionality [27]. This synergistic effect between the matrix and the anthocyanins could be responsible for the enhanced antioxidant and bioactive properties observed.

#### 2.2.1. O-H Stretching Region (3200–3600 cm^−1^)

The O-H stretching region, commonly associated with hydroxyl groups in phenolic compounds, showed significant changes upon encapsulation (Table 2 and Table 3). In the non-microencapsulated samples of *V. floribundum* and *R. glaucus*, the O-H stretching peaks were pronounced and of high intensity, reflecting the presence of free hydroxyl groups. In contrast, the microencapsulated samples exhibited reduced peak intensities, suggesting that these groups were less exposed due to their interaction with the maltodextrin matrix. This retention of key functional groups, including hydroxyl, alkyl, and phenolic groups, indicates that the encapsulation process does not significantly alter the chemical structure of the anthocyanins, which is critical for maintaining their antioxidant properties and potential applications [28,29]. The protective effect is crucial for enhancing the stability and longevity of anthocyanins in functional applications, such as food and nutraceutical products.

#### 2.2.2. C-H Stretching Region (2800–3000 cm^−1^)

Distinct differences were observed in the C-H stretching region between the non-microencapsulated and microencapsulated forms of both berries, as detailed in Table 2 and Table 3. In the non-microencapsulated samples, the C-H peaks were either absent or of low intensity, indicating minimal presence in the anthocyanins. However, medium-intensity peaks were observed in the microencapsulated samples, attributed to maltodextrin. This confirms the presence of the encapsulating agent and supports the successful incorporation of maltodextrin into the anthocyanin structure, reinforcing the effectiveness of the encapsulation process.

#### 2.2.3. C=O Stretching (1700–1750 cm^−1^)

Carbonyl stretching peaks, associated with C=O groups in anthocyanins, showed high intensity in the non-microencapsulated forms of both berries, as seen in Table 2 and Table 3. These peaks were notably reduced in the microencapsulated samples, suggesting interactions between the carbonyl groups and the maltodextrin matrix, likely through hydrogen bonding. This interaction partially shields the carbonyl groups, contributing to enhanced stability by reducing the reactivity of these groups and preventing degradation [30]. The reduction in exposure of these reactive groups supports the protective role of encapsulation, which may prevent degradation processes common in non-encapsulated anthocyanins.

#### 2.2.4. Aromatic C=C Stretching (1500–1600 cm^−1^)

The aromatic C=C stretching peaks, indicative of the aromatic rings in anthocyanins, were medium in intensity in the non-microencapsulated samples (Table 2 and Table 3). Encapsulation led to a reduction in the intensity of these peaks, indicating that the aromatic rings are partially protected by the maltodextrin matrix. This decreased exposure suggests that encapsulation decreases the exposure of these structurally significant components, thereby preserving the integrity of the anthocyanins during storage and use, which is essential for maintaining their antioxidant properties.

#### 2.2.5. C-O and C-O-C Stretching (1000–1300 cm^−1^)

Both non-microencapsulated and microencapsulated samples exhibited C-O and C-O-C stretching peaks, with the microencapsulated samples showing lower intensities (Table 2 and Table 3). These differences suggest interactions between the anthocyanins and the encapsulating maltodextrin matrix, affecting the ether and glycosidic bonds and enhancing the stabilization of these functional groups. The introduction of new peaks in the microencapsulated spectra, corresponding to the encapsulating materials, suggests effective protection and stabilization of anthocyanins. This encapsulation likely enhances the stability of anthocyanins, protecting them from degradation due to environmental stressors such as pH, temperature, and light exposure [25].

#### 2.2.6. C-O-C Stretching of Polysaccharides (1027 cm^−1^)

A distinct C-O-C stretching peak, associated with polysaccharides and maltodextrin, was absent in the non-microencapsulated samples but appeared prominently in the microencapsulated forms of both berry extracts (Table 2 and Table 3). This peak serves as direct evidence of successful encapsulation, highlighting the presence and incorporation of maltodextrin within the anthocyanin microspheres.

The combined FTIR analysis, as detailed in Table 2 and Table 3 and Appendix A, demonstrates that maltodextrin encapsulation significantly alters the structural environment of anthocyanins in both *R. glaucus* and *V. floribundum*. The reduction in the intensity of key anthocyanin peaks and the appearance of new peaks related to maltodextrin confirm the successful encapsulation process. The scientific literature corroborates these findings, highlighting that encapsulation can introduce new vibrational modes and mask or shift original anthocyanin peaks. The observed shifts in O-H stretching and masking of aromatic C=C stretching further indicate interactions between anthocyanins and encapsulating materials, potentially leading to improved stability and controlled release [30]. Encapsulation effectively shields anthocyanins from environmental degradation, supporting their functional properties and making it a valuable technique for improving the bioactive potential of berry extracts in food and nutraceutical applications. Future studies, including stability tests and release profiles, will further validate the benefits and applications of microencapsulated anthocyanins in various industries.

With these results, our research provides evidence that maltodextrin enables sustained release of the biomolecules presenting practical applications in both functional food and pharmaceutical contexts.

### 2.3. Morphological Analysis by Scanning Electron Microscope (SEM)

Morphological analysis using SEM is a well-established technique widely used to characterize microencapsulated spheres, such as those prepared with maltodextrin using spray-drying methods. SEM imaging provides valuable insights into particle size, shape, and surface morphology, which are essential for evaluating encapsulation efficiency, stability, and potential applications in food and pharmaceutical formulations [31,32].

SEM analysis of microencapsulated spheres, as seen in the studies of both *V. floribundum* and *Rubus glaucus*, reveals that the spray-drying process produces predominantly spherical particles with varying degrees of uniformity and occasional agglomeration. The slightly narrower particle size distribution of *R. glaucus* compared to *V. floribundum* could be attributed to differences in the initial anthocyanin composition and viscosity of the extracts, which influence droplet formation during the spray-drying process.

In the case of *R. glaucus*, the SEM images (Figure 1) captured at 1.00 k× magnification (with a scale bar of 50 μm) showed mostly aggregated spherical particles with smooth surfaces, indicative of successful encapsulation. However, some particles appear clustered or fused, likely reflecting the influence of suboptimal drying conditions. Similarly, the SEM analysis of *V. floribundum* spheres (Figure 2) also highlights a heterogeneous distribution of particle sizes and shapes. Predominantly spherical particles are observed, but some irregular fragments and broken or agglomerated particles suggest variations in droplet sizes or drying rates. The differences in particle size and shape observed in the SEM analysis may influence the efficacy of microencapsulation, as smaller and more uniform particles tend to enhance encapsulation efficiency, stability, and controlled release, while larger or irregular particles could reduce these effects [33].

Quantitative analysis using the FIJI software (version 2.9.0) for particle size distribution of both samples confirmed a positively skewed histogram. For *V. floribundum*, the histogram (Figure 3) shows a peak at 1.28 μm, indicating the most frequent particle size, while the mean particle size is 2.65 μm with a standard deviation of 1.82 μm. The particle sizes range from as small as 0.27 μm to over 9 μm, but larger particles are less frequent. The cumulative size distribution curve indicates that approximately 40% of the particles are smaller than 1.28 μm, with a steep rise between 1.28 and 3.30 μm, confirming that a majority fall within this range. These results suggest that the encapsulation process predominantly produces small to medium-sized particles with moderate variability in sizes.

Similarly, the particle size distribution of *R. glaucus* (Figure 4) shows a mode at 1.09 μm, with a mean size of 2.21 μm and a standard deviation of 1.70 μm. The range extends from 0.3 μm to over 10 μm, with most particles below 2 μm. The cumulative size distribution indicates that about 80% of the particles are smaller than 3 μm, consistent with the presence of fine particles, while a plateau beyond 5 μm suggests minimal larger particles. This trend parallels that of *V. floribundum*, demonstrating that both encapsulation processes yield predominantly fine particles, though with slight differences in the size ranges and distributions. The observed differences in particle morphology between *R. glaucus* and *V. floribundum* may be due to variations in the chemical composition and viscosity of the extracts, which affect droplet formation during the spray-drying process. These inherent differences between the species likely contribute to the variations in particle size and shape.

Factors like feed concentration, drying temperature, and atomization rate can be optimized to minimize aggregation and improve the uniformity of the encapsulated product [34]. Based on similar studies, adjusting the feed concentration to 20–25% solid content and maintaining inlet temperatures between 140 and 160 °C have been shown to optimize encapsulation efficiency and particle size uniformity [35]. Future studies could optimize spray-drying conditions by further adjusting parameters such as feed concentration, drying temperature, and atomization rate. Additionally, fine-tuning the atomization rate could help produce more consistent particles, leading to improved overall encapsulation quality. Achieving a more consistent particle size distribution would enhance product quality, particularly for applications requiring rapid dissolution or high surface area [36,37].

Overall, SEM analysis proves invaluable for understanding the morphology and size distribution of microencapsulated particles. These findings are essential for evaluating the encapsulation efficiency and stability of the microencapsulated product. Smaller particle sizes are generally advantageous for applications requiring rapid dissolution or dispersion, such as in food or pharmaceutical formulations [38,39]. The relatively narrow size distribution and predominance of fine particles indicate effective spray-drying parameters, though there remains potential for process optimization. Reducing the presence of larger or fragmented particles could enhance product uniformity. Adjustments to feed concentration, drying temperature, or atomization rate might be necessary to improve overall encapsulation quality and particle size consistency [40,41].

### 2.4. Antibacterial Activity

The antibacterial activity of microencapsulated anthocyanin extracts from *R. glaucus* and *V. floribundum*, prepared via spray-drying with maltodextrin, was evaluated against a panel of Gram-positive and Gram-negative bacterial strains using the agar-well diffusion method. The results, shown in Figure 5, Figure 6 and Figure 7, demonstrate a concentration-dependent antibacterial effect for both extracts, with variations depending on the bacterial strain tested. Notably, maltodextrin at the maximum concentration tested (550 mg/mL) did not show any antibacterial effect, which is consistent with previous reports [42], highlighting that the growth inhibition seen for the microencapsulated extracts is derived from the extract itself. The observed particle morphology, particularly the spherical shape and smooth surface, likely contributes to the improved stability of anthocyanins and controlled release, enhancing the antioxidant and antibacterial efficacy of the microencapsulated extracts.

Figure 5 shows the antimicrobial efficacy of the microencapsulated *R. glaucus* extract against various bacterial strains, with inhibition zones (in mm) and their corresponding standard deviations displayed as a heatmap. At the highest concentration tested (479.33 mg/mL), the extract exhibits significant inhibitory effects against *Pseudomonas aeruginosa* (14.33 ± 0.58 mm), *Enterococcus faecalis* (15.67 ± 1.15 mm), *Bacillus cereus* (15.07 ± 0.12 mm), and *Listeria monocytogenes* (20.33 ± 0.58 mm).

Figure 6 presents the antimicrobial activity of the microencapsulated *V. floribundum* extract, showing inhibition zones across different concentrations and bacterial strains, along with their standard deviations. At the highest concentration tested (542 mg/mL), the *V. floribundum* extract exhibits inhibitory effects against *P. aeruginosa* (10.83 ± 0.29 mm), *E. faecalis* (12.50 ± 0.87 mm), *B. cereus* (13.33 ± 0.58 mm), and *L. monocytogenes* (12.67 ± 1.15 mm). Although the inhibition zones for *P. aeruginosa* (10.83 ± 0.29 mm) and *E. faecalis* (12.50 ± 0.87 mm) are greater than those produced by the corresponding antibiotics (8.83 ± 0.76 mm and 10.67 ± 0.58 mm, respectively), these results must be considered in the context of the much higher concentration of the extract. The *V. floribundum* extract demonstrates moderate activity against *L. monocytogenes* (12.67 ± 1.15 mm), which, while less effective than the *R. glaucus* extract, still exceeds the inhibition of carbenicillin.

Figure 7 provides a clustered bar chart comparing the Minimum Inhibitory Concentration (MIC) values of both microencapsulated extracts against the tested bacterial strains. The MIC data indicate that the *R. glaucus* extract generally has lower MIC values across all tested bacteria, suggesting higher antimicrobial potency compared to *V. floribundum*. For example, the MIC of *R. glaucus* against *E. faecalis* is 119.83 mg/mL, compared to 135.5 mg/mL for *V. floribundum*. Similarly, the MIC against *P. aeruginosa* is 239.66 mg/mL for *R. glaucus* versus 271 mg/mL for *V. floribundum*. The lowest MIC values are observed for *B. cereus*, with MICs below 29.95 mg/mL for *R. glaucus* and below 33.87 mg/mL for *V. floribundum*, highlighting their strong inhibitory effect against this Gram-positive bacterium, which is aligned to previous research [43]. The results indicate that the *R. glaucus* extract generally has lower MIC values, suggesting it possesses higher antimicrobial potency compared to the *V. floribundum* extract.

The comparison in Figure 7 also shows that both extracts have relatively high MIC values against *P. aeruginosa* (239.66 mg/mL for *R. glaucus* and 271 mg/mL for *V. floribundum*), indicating that this strain is less susceptible to the microencapsulated anthocyanins. However, the *R. glaucus* extract consistently demonstrates a more effective inhibitory effect, with lower MIC values overall, reinforcing its potential as a more potent antimicrobial agent of the two extracts studied. In prior evaluations of the extracts without microencapsulation, consistently lower MIC values were also observed for *R. glaucus* compared to *V. floribundum* [9].

These findings suggest that while both microencapsulated extracts exhibit promising antimicrobial activity, particularly at high concentrations, their efficacy compared to antibiotics is limited by the much higher concentrations required to achieve similar effects. Tetracycline and carbenicillin, as pure antibiotics, achieve bacterial inhibition at significantly lower concentrations due to their high purity and specific mode of action; these findings are also consistent with the previous literature. For example, anthocyanins generally require higher concentrations to achieve comparable effects to antibiotics like ciprofloxacin, as their mode of action is less targeted and involves multiple biochemical pathways [44]. The data in Figure 5, Figure 6 and Figure 7 emphasize the need to optimize the bioactive components in these extracts, potentially through purification or synergistic combinations with other antimicrobials, to enhance their potency and reduce the required concentrations for effective use.

The enhanced antibacterial activity of the microencapsulated anthocyanins may be attributed to their ability to disrupt bacterial cell membranes, increasing permeability and leading to cell lysis. This is consistent with other studies demonstrating the membrane-disruptive effects of anthocyanins [45,46]. This mechanism is more effective against Gram-positive bacteria due to the absence of an outer membrane, which contrasts with Gram-negative bacteria that have a protective lipopolysaccharide layer. Such differences in membrane structure partly explains the higher MICs observed against Gram-negative strains in our study and similar findings previously reported [47].

Overall, these results underscore the potential of *R. glaucus* and *V. floribundum* extracts as alternative or complementary antimicrobial agents, particularly in settings where natural products are preferred or where antibiotic resistance is a concern. Future studies should focus on refining the extraction and encapsulation processes, increasing the concentration of active compounds, and investigating synergistic effects with conventional antibiotics to maximize therapeutic efficacy.

### 2.5. Antitumor Activity

*R. glaucus* and *V. floribundum* are important sources of bioactive compounds, including anthocyanins and polyphenols, which are known for their ability to eliminate free radicals that can damage cells, strengthen the immune system, and potentially reduce the risk or incidence of cancer [48]. Dose–response curves were employed to calculate the inhibitory concentration values (IC_50_), which indicate the concentration of a substance required to inhibit cell proliferation by 50% (Table 4).

These values serve as a key indicator of the efficacy of the microencapsulated extracts, with lower IC_50_ values indicating higher potency against specific cell lines. The results show a clear difference in the antitumor efficacy between *R. glaucus* and *V. floribundum* (Appendix A). The IC_50_ values for the microencapsulated *R. glaucus* extract range from 3.07 to 5.03 mg/mL, suggesting a higher potency in inhibiting tumor cell proliferation compared to *V. floribundum*, which has IC_50_ values ranging from 8.15 to 15.59 mg/mL. Corresponding concentrations of anthocyanins are indicated for the IC_50_ of each microencapsulation (Appendix A), highlighting the role of these bioactive compounds in contributing to the observed cytotoxic effects. Despite *V. floribundum*’s higher antioxidant potential (Table 1), *R. glaucus* proves more potent at lower concentrations. This disparity implies that antioxidant activity alone does not correlate with antitumoral potency; factors such as bioavailability or specific molecular interactions in *R. glaucus* likely enhance its tumor inhibition. In prior evaluations of the extracts without microencapsulation, consistently lower IC_50_ values were also observed for *R. glaucus* compared to *V. floribundum*, except in MDA-MB-231 cells [9]. Additionally, maltodextrin reflected negligible toxic effects on cells (Appendix A). This confirms that maltodextrin alone had no significant impact on cell viability, ensuring the observed effects are due to the active compounds and not the encapsulation material. Conversely, for non-tumor cells (NIH3T3), *R. glaucus* exhibited an IC_50_ of 4.32 mg/mL, while *V. floribundum* showed a significantly higher IC_50_ of 10.44 mg/mL, potentially leading to better therapeutic outcomes.

Based on the IC_50_ values for non-tumor cells compared to those for tumor cells, the therapeutic indexes (TI) were calculated (Figure 8). A higher TI reflects a more favorable therapeutic profile, characterized by an effective inhibition of tumor cells with minimal impact on healthy cells. The TI values for *R. glaucus* (0.57–1.40) and *V. floribundum* (0.70–1.30) reveal that both microencapsulations exhibit similar profiles, suggesting a comparable therapeutic window. The highest TI for *V. floribundum* is observed in breast cancer cells (MDA-MB-231), while *R. glaucus* shows promising TI values in melanoma (SKMEL-103). However, both microencapsulations exhibit relatively low TI values across the tested cell lines. This suggests a narrow therapeutic window and raises concerns about the potential cytotoxicity to healthy cells alongside their antitumor effects. In summary, while the TI values indicate that further optimization is necessary to enhance the therapeutic potential, *R. glaucus* shows lower IC_50_ values for tumor cells, indicating higher potency.

Microencapsulation has proven to be a promising strategy to enhance the stability and efficacy of bioactive compounds, protecting them from degradation by controlling their reactivity, photosensitivity, and durability before they reach the tumor site while allowing for controlled release to ensure sustained therapeutic effects over time [49]. Importantly, this strategy not only preserves the compound’s potency but also enhances bioavailability by improving absorption and cellular uptake, as demonstrated in studies on anthocyanins, which are crucial for the antitumor effects of berries [50,51]. For instance, A. Kazan et al. reported that the encapsulation of *V. floribundum* extracts with chitosan resulted in lower IC_50_ values in an osteosarcoma cell line, although similar benefits were not observed in the other tumor cell lines tested [52]. Microencapsulation of different fruit extracts, such as tamarillo, has shown dose-dependent cytotoxic effects on various cancer cell lines, with encapsulation conditions significantly influencing IC_50_ values and thereby impacting anticancer efficacy due to variability in phytochemical retention [53]. Additionally, the microencapsulation of a *Gratiola officinalis* extract has been shown to significantly reduce viability in breast cancer cells without inducing the formation of autophagosomes, which are indicators of antitumor resistance [54]. While microencapsulation can enhance cellular uptake, its effects are more pronounced with longer incubation times and in vivo settings. The choice of matrix and encapsulation techniques may significantly influence the antitumor efficacy of the extracts [55,56].

### 2.6. Anti-Inflammatory Activity

Inflammation is a primary defense response to harmful stimuli, and plant extracts have been shown to inhibit pro-inflammatory mediators involved in this process [57]. The anti-inflammatory potential of microencapsulated *R. glaucus* and *V. floribundum* extracts was assessed by measuring nitric oxide (NO) production in LPS-stimulated RAW264.7 cells (Figure 9). Pretreatment with 2.5 mg/mL of microencapsulated extracts for 4 h before LPS stimulation reduced NO production to 74.9% for *V. floribundum* and 71% for *R. glaucus* compared to the LPS-only control. This reduction indicates the anti-inflammatory properties of these extracts, as excessive NO production is a recognized marker of inflammation [58]. Importantly, cell viability assays confirmed that microencapsulation did not compromise cell health, with viability exceeding 90% in all treated groups, thereby ensuring that the reduction in NO was due to anti-inflammatory activity rather than cell death. Furthermore, the effectiveness of the extracts was demonstrated when compared to dexamethasone, a standard anti-inflammatory drug. At lower concentrations, microencapsulated *R. glaucus* showed a NO reduction of up to 86.1%, while *V. floribundum* did not show positive results. Higher concentrations could not be tested due to cell growth inhibition. Additionally, maltodextrin alone reduced NO production to 91% of the control.

The anti-inflammatory effects are attributed to bioactive compounds such as anthocyanins and polyphenols, which are known to scavenge free radicals and inhibit inflammatory pathways [59]. Microencapsulation likely preserved and enhanced the bioactivity and stability of these compounds, consistent with existing research on their benefits in reducing inflammation and oxidative stress [60]. The significant reduction in NO production observed in this study aligns with the documented anti-inflammatory effects of these bioactive molecules and extracts [61]. Overall, our results suggest that microencapsulation enhances the anti-inflammatory effects of the extracts [9].

## 3. Methods

### 3.1. Plant Material

Berries were acquired from a local market in Ambato, Ecuador, and transported immediately to the laboratory, where the fruits were visually selected for uniformity in size, degree of maturity, and absence of defects. Then, the fruits were washed with tap water, disinfected with a 100 ppm chlorine solution; the fruits were stored at −20 °C and freeze-dried using a Labconco (Kansas City, MO, USA) FreeZone^®^ 6 Liter Benchtop Freeze Dryer. The primary drying phase was carried out under vacuum at a pressure of 50–150 mTorr and a chamber temperature of −50 to −30 °C, allowing the sublimation of frozen water. This phase removed approximately 90–95% of the water content. The secondary drying phase involved raising the temperature to 20–40 °C to remove any remaining bound water, reducing the moisture content to 1–4%. The entire freeze-drying process lasted for 72 h. The dried samples were then ground into a fine powder using an electric lab blender. The sieve N35 (500 μm) was used. The samples were stored at −20 °C and utilized for the experiments in this study.

### 3.2. Anthocyanin Extraction

The ground material was mixed with a solution of twenty times its volume consisting of 96% ethanol and 1.5 M HCl in an 85:15 *v*/*v* ratio. This mixture was added to a stirrer tank and subjected to extraction at 70 °C for 60 min. The solid components were separated from the mixture using an Andreas Hettich GmbH & Co (Tuttlingen, Germany) Rotine 380 centrifuge at 6000 RPM for 15 min at 4 °C to ensure efficient separation and prevent degradation of the bioactive compounds. The extract was then transferred to a 500 mL flask, continuously stirred, and subsequently subjected to rotary evaporation in a water bath at 70 °C for 2 h under vacuum to remove the solvent [62].

### 3.3. Quantification of Total Anthocyanins and Phenolic Compounds

The total anthocyanin content in the lyophilized extract was determined using the differential pH method described by Lee et al. 2005 [63]. This method employs two buffer systems: potassium chloride at pH 1.0 and 0.025 mol/L, and sodium acetate at pH 4.5 and 0.4 mol/L. The absorbance of the fruit extract was measured at 515 nm and 700 nm using a Thermo Scientific (Waltham, MA, USA) Genesys 10S UV-Vis spectrophotometer in both buffer solutions. The absorbance difference was calculated using the formulas:(1)$$A=\left(A510-A700\right)pH1-(A510-A700)pH4.5$$
(2)$$TA=\frac{A\times MW\times DF\times 100}{\varepsilon \times 1}$$
where A = absorbance; MW = molecular weight; DF = dilution factor; and ε = extinction coefficient. Data were calculated using the MW (449.20) and the extinction coefficient for cyanidin-3-glucoside (29,600) and expressed as mg cyanidin/100 g

The total phenolic content (TPC) was determined using the Folin–Ciocalteu reagent with catechin as the standard [64]. Briefly, 1 mL of Folin–Ciocalteu reagent (FCR) and 0.8 mL of 7.5% sodium carbonate were combined with 0.2 mL of the sample extract. The mixture was shaken and incubated at room temperature for 30 min. The absorbance was then measured at 765 nm using a Thermo Scientific (Waltham, MA, USA) Genesys 10S UV-Vis spectrophotometer. A calibration curve with gallic acid standards ranging from 50 to 250 μg/mL was used to quantify the TPC, expressed as micrograms of gallic acid equivalents (GAE) per gram of sample. Each measurement was performed in triplicate.

### 3.4. Microencapsulation

Carrier agents for spray-drying, specifically maltodextrin (10–20 DE), were obtained from Roig Farma, Terrassa, Spain. The carrier agent (maltodextrin 10–20 DE) was mixed with the anthocyanin extract (27% solid content) in a ratio of 20:80 anthocyanins to maltodextrin, equating to 20 g of anthocyanins. To this mixture, 280 mL of distilled water was added, and it was stirred to homogeneity using an RH Digital (Neuss, Germany) Ika mixer.

The feed mixtures were spray-dried using a BÜCHI Labortechnik AG (Flawil, Switzerland) Büchi Mini Spray Dryer B-290 at air inlet/outlet temperatures of 150/90 °C. The resulting powder was collected in a collection flask. The pigment powders were stored in HDPE-aluminum bags at 15–25 °C.

### 3.5. Characterization by FTIR

Fourier Transform Infrared Spectroscopy (FTIR) was used to analyze the functional groups present in the microcapsules. The FTIR spectra were recorded on a JASCO FTIR 4100 spectrometer (Tokyo, Japan) using the ATR method, covering the wave number range of 4000–500 cm^−1^ with a resolution of 4 cm^−1^ over 36 scans.

### 3.6. Morphological Analysis by Scanning Electron Microscope (SEM)

Each sample was mounted on a stub for electron microscopy using conductive double-sided carbon tape. The samples were then coated with approximately 20 nm of conductive gold (99.99% purity) for 60 s using a Quorum (San Jose, CA, USA) Q150R ES sputtering evaporator. The micrographs were taken with a TESCAN MIRA 3 Scanning Electron Microscope (SEM) using the Secondary Electron (SE) detector.

The images were analyzed using the biological image analysis platform Fiji (version 2.9.0) [65]. Area data were obtained from the images using Fiji. The diameter of each sphere was calculated using the formula:(3)$$D=\frac{2\surd A}{\pi }$$
where D is the diameter and A is the area obtained from Fiji.

### 3.7. Antibacterial Assay

The microencapsulated extract’s antibacterial activity was evaluated against the Gram-positive bacteria *Staphylococcus aureus* ATCC 25923, *Enterococcus faecalis* ATCC 29212, *Bacillus cereus*; *Listeria monocytogenes* ATCC 13932; and the Gram-negative bacteria *Pseudomonas aeruginosa* ATCC 27853, *Escherichia coli* ATCC 25922, *Salmonella enterica* ATCC 14028, *Klebsiella ozaenae*, and *Enterobacter gergoviae*.

Stock solutions of the microencapsulated anthocyanins were prepared by independently dissolving 479 mg of microencapsulated *R. glaucus* extract powder and 542 mg of microencapsulated *V. floribundum* extract powder in 1 mL of sterile distilled water. These values correspond to the maximum solubility of the microencapsulated samples. Importantly, the stock concentrations of both microencapsulated samples equate to 0.667 mg of anthocyanins alone. Maltodextrin alone was used as negative control at a final concentration of 550 mg/mL. Tetracycline (10 μg/mL) was used as a control to inhibit *Pseudomonas aeruginosa*, while carbenicillin (100 μg/mL) served as a control against *Enterococcus faecalis*, *Bacillus cereus*, and *Listeria monocytogenes*. These antibiotics were chosen due to their known effectiveness against the corresponding bacterial strains [66].

Antibacterial activity assessment was conducted using the agar-well diffusion method [67]. For this purpose, the bacterial inoculum was spread on Mueller–Hinton agar, and wells of 6 mm diameter were punched. Then, each well was filled with either 80 μL of the microencapsulated extract stock or the controls. The plates were incubated overnight at 37 °C. Lastly, the antibacterial potential was determined by measuring the diameter (in mm) of the inhibition area and compared to the diameter of the antibiotic at the standard concentration. Each assay was performed in triplicate.

Once the bacterial strains that were inhibited by the microencapsulated extracts at their maximum concentrations (542 mg/mL and 479 mg/mL) were determined, a second round of analysis by well-agar diffusion was conducted to narrow down the inhibitory concentration. Each well was filled with 80 μL of a microencapsulated extract dilution. For this purpose, the berries stock solutions were serially diluted up to four times to yield concentrations ranging from 33.88 to 542 mg/mL and 29.95–479 mg/mL, respectively. The plates were incubated overnight at 37 °C, after which the inhibition area was measured and reported in mm. Each assay consisted of three technical and three biological replicates.

### 3.8. Determination of Minimum Inhibitory Concentration (MIC)

The Minimum Inhibitory Concentration (MIC) for each bacterial strain was determined through a series of dilutions of the microencapsulated extracts from *Rubus glaucus* and *Vaccinium floribundum*. Stock solutions were prepared at the maximum solubility levels, 479 mg/mL for *R. glaucus* and 542 mg/mL for *V. floribundum;* these values correspond to the maximum solubility levels of the microencapsulated samples. These solutions were then serially diluted four times, resulting in concentrations ranging from 29.95 to 479 mg/mL for *R. glaucus* and from 33.88 to 542 mg/mL for *V. floribundum*. The diluted samples were tested against bacterial strains using the agar-well diffusion method, and the MIC was defined as the lowest concentration at which no visible bacterial growth was observed after overnight incubation at 37 °C.

### 3.9. Antitumor Activity

Four tumoral cell lines, MDA-MB-231 (human breast adenocarcinoma, ATCC No. HTB-26), SK-MEL-3 (human melanoma, ATCC No. HTB-69), HCT116 (human breast adenocarcinoma, ATCC No. CCL-247), and HT29 (human colorectal adenocarcinoma, ATCC No. HTB-38), and one non-tumoral cell line, NIH3T3 (mouse NIH/Swiss embryo fibroblasts), were obtained from ATCC and cultured in Dulbecco’s modified eagle medium (DMEM/F12) (Corning, Manassas, VA, USA) supplemented with 10% fetal bovine serum (FBS) (Eurobio, Les Ulis, France) and 1% penicillin/streptomycin (Thermo Fisher Scientific, Gibco, Miami, FL, USA). All cell lines were maintained at 37 °C in a ESCO, (Egaa, Denmark) CO_2_ incubator model CCL-170T-9, humidified atmosphere with 5% CO_2_. To analyze the effect of the extracts on cell proliferation, cells were seeded at a density of 1 × 10^4^ cells/well in 100 μL in 96-well plates (clear TC-treated polystyrene, flat bottom, Corning). Cells were incubated for 72 h with 100 μL of the microencapsulated extracts of *R. glaucus* and *V. floribundum* at 0.6–42 mg/mL of microencapsulation (0.78–50 μg/mL final concentrations of anthocyanins). The effect of maltodextrin, the encapsulating agent, was also evaluated using the same method utilizing 100 μL of the compound at 0.6–167 mg/mL final concentrations. Following the specified incubation period, the thiazolyl blue tetrazolium bromide (MTT) assay was performed as previously described [68,69]. In summary, 10 μL of MTT solution (5 mg/mL) was added to each well. After allowing 1–2 h for incubation in a humidified environment, the media was aspirated, and 50 μL of DMSO was added to each well to dissolve the formazan crystals. The mixture was agitated for 5 min before measuring the absorbance at 540 nm (top monochromator probe 2.0 mm) using a Cytation5 multimode detection system (BioTek, Winooski, VT, USA). Each data point was obtained from quadruplicate samples, and the experiment was replicated at least four times. Cells with media alone were used as negative controls and chemotherapeutic agent cisplatin (50 μg/mL) was used as a positive control in all experimental designs. To determine the concentration of the compound required to inhibit 50% of cell proliferation (IC_50_), dose–response curves were generated in the GraphPad Prism 10.2 software (GraphPad Software, San Diego, CA, USA) using the control group of untreated cells as 100% cell proliferation.

### 3.10. Anti-Inflammatory Assay

RAW264.7 murine macrophage cells were obtained from ATCC and cultured in RPMI (Corning, Manassas, VA, USA) supplemented with 10% fetal bovine serum (FBS) (Eurobio, Les Ulis, France) and 1% penicillin/streptomycin (Thermo Fisher Scientific, Gibco, Miami, FL, USA). Cells were maintained at 37 °C in a humidified atmosphere with 5% CO_2_ (CO_2_ incubator, ESCO, model CCL-170T-9, Denmark). To analyze the anti-inflammatory effects of *R. glaucus* and *V. floribundum*, RAW264.7 cells were cultured in 24-well plates (clear TC-treated polystyrene, flat bottom, Corning) at a concentration of 4 × 10^5^ cells per well. The macrophage cells were first exposed for 4 h to the microencapsulated extracts at concentrations 1.5–2.5 mg/mL before being stimulated for 18 h with 1 μg/mL of lipopolysaccharide (LPS). Also, dexamethasone (DEX) (0.5 μg/mL) and maltodextrin (1.5–2.5 mg/mL) were included in the assay. After the treatment, supernatants were collected to assess nitric oxide (NO) production using the Griess reagent system as previously described [70]. Specifically, 50 μL of Griess reagent was added to 50 μL of each supernatant in a 96-well plate (clear, flat bottom, Corning), and the mixture was incubated for 10 min at room temperature in the dark. NO production was indicated by the intensity of color produced from the reaction between the culture medium and the Griess reagent. Absorbance was measured at 540 nm (top monochromator probe 2.0 mm) using a Cytation5 multimode detection system with a built-in spectrophotometer and programmable agitation function for uniform mixing (BioTek, Winooski, VT, USA). Cells stimulated only with LPS were used as positive control for inflammation (NO production), LPS + dexamethasone was used as the positive control for anti-inflammatory activity, and cells incubated only with media were used as negative control. The concentrations of the compounds used for the assay were chosen based on their IC_50_ values and did not significantly affect cell growth. The assay was performed at least in triplicate.

### 3.11. Antioxidant Assay

To determine the antioxidant activity of microencapsulated extracts, maltodextrin was first removed from the samples. From each microencapsulated extract, 1.5 g was dissolved in 12.5 mL of sterile distilled water (stock solution). Then, 1 mL of the stock was mixed with 9 mL of 100% methanol and incubated with rotation for 30 min at RT. The samples were centrifuged at 2800 rcf for 10 min. The supernatant was collected, and the process was repeated twice with the pellet. The final concentration of the extract without maltodextrin was adjusted to 6 mg/mL with 100% methanol and then used to assess antioxidant activity.

Antioxidant activity was then assessed by the 2,2-diphenyl-1-picrylhydrazyl (DPPH) assay according to Barba-Ostria et al. (2024) [9]. The following formula was used to calculate the %DPPH scavenging activity:$$\%\;DPPH\;scavenging=100*\frac{A_{sample+DPPH}-A_{sample\;blank}}{A_{DPPH}-A_{solvent}}$$

IC_50_ values were calculated using GraphPad Prism version 10.2 (GraphPad Software, Corp., Boston, MA, USA). All the results were given as a mean ± standard deviation (SD) of experiments conducted at least in triplicate.

### 3.12. Statistical Analysis

Independent two-sample *t*-tests were used to compare total polyphenol content (TPC), anthocyanin content, and antioxidant activity (IC_50_ values) between the two extracts. Additionally, the antioxidant activity of both extracts was compared to a control (ascorbic acid, IC_50_: 3.84 ± 0.92 μg/mL). The mean and standard deviation of each parameter were used for the comparisons, with a significance level of *p* < 0.05. Results were considered statistically significant if *p* < 0.001, indicating substantial differences between the extracts and the control. Statistical analysis of antitumor activity was performed using two-way ANOVA to compare the IC_50_ values of the two microencapsulated extracts for each cell line (significant differences, *p* < 0.001) and unpaired *t*-test was conducted to assess the means within each group (significant differences, *p* < 0.001).

## 4. Conclusions

This study provides comprehensive insights into the bioactive properties of microencapsulated anthocyanins derived from *V. floribundum* (Andean blueberry) and *R. glaucus* (Andean blackberry). The use of microencapsulation, a technique involving the entrapment of bioactive compounds within a protective matrix, has proven effective in enhancing the stability, bioavailability, and biological activity of these anthocyanin-rich extracts.

The microencapsulated extracts exhibited significant antioxidant activity, as demonstrated by their ability to scavenge free radicals in the DPPH assay. Notably, *V. floribundum* showed higher polyphenol and anthocyanin contents, which correlate with its superior antioxidant activity seen.

The antibacterial properties of the microencapsulated extracts were evaluated against a panel of Gram-positive and Gram-negative bacterial strains. Both extracts demonstrated a concentration-dependent antimicrobial effect, with *R. glaucus* generally showing higher potency, as indicated by lower MIC values across several bacterial strains. These results suggest that these microencapsulated extracts could serve as potential natural alternatives or complements to conventional antibiotics, particularly in contexts where resistance to standard treatments is a concern. However, the need for higher concentrations compared to pure antibiotics highlights the necessity for further optimization to enhance their efficacy.

The study also explored the cytotoxic and antitumor activities of the microencapsulated extracts. The microencapsulated *R. glaucus* extract demonstrated greater potency in inhibiting tumor cell proliferation, as evidenced by lower IC_50_ values across various cancer cell lines, compared to *V. floribundum*. However, *V. floribundum* exhibited a more favorable therapeutic index (TI), indicating a safer profile due to its selective toxicity towards cancer cells while sparing normal cells. These findings suggest that while both extracts hold promise for anticancer applications, *V. floribundum* may offer a safer therapeutic option, particularly when minimizing cytotoxicity to healthy cells is a priority.

In addition to their antioxidant, antibacterial, and antitumor activities, the microencapsulated extracts demonstrated notable anti-inflammatory effects. The reduction in nitric oxide (NO) production in LPS-stimulated macrophages indicates that both extracts possess significant anti-inflammatory properties. *R. glaucus* showed a greater reduction in NO levels, further reinforcing its potential as a potent anti-inflammatory agent. Importantly, the preservation of cell viability throughout these assays confirmed that the observed anti-inflammatory effects were due to the bioactive compounds themselves rather than cytotoxicity.

Overall, this study highlights the effectiveness of microencapsulation in preserving and enhancing the functional properties of anthocyanins, which are critical for their health-promoting effects. The microencapsulated extracts of *V. floribundum* and *R. glaucus* exhibit a broad spectrum of biological activities, including antioxidant, antibacterial, anticancer, and anti-inflammatory effects. These findings suggest that microencapsulated anthocyanins could be used as natural preservatives in food products, providing an alternative to synthetic preservatives. Furthermore, their potential as therapeutic agents against antibiotic-resistant bacteria warrants further investigation. Future research should focus on optimizing encapsulation conditions, exploring synergistic combinations with other bioactives, and conducting in vivo studies to fully realize the therapeutic potential of these natural compounds.

## Figures and Tables

**Figure 1 molecules-29-05504-f001:**
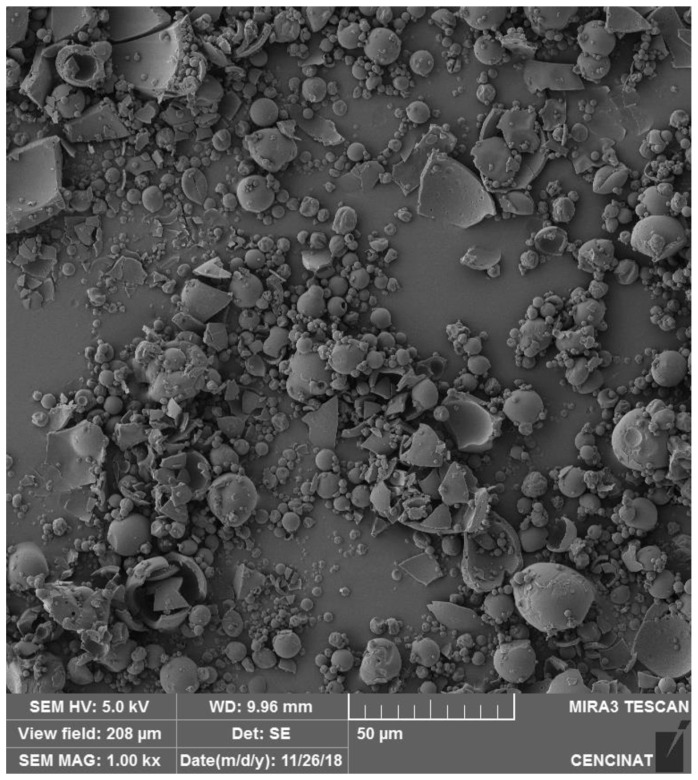
Morphological characterization of *R. glaucus* anthocyanins microencapsulated particles. Scanning Electron Microscopy (SEM) image of *R. glaucus* microencapsulated particles produced by spray-drying. The image was captured using a TESCAN (Brno, Czech Republic) MIRA 3 SEM at 1.00 k× magnification with a 50 μm scale bar.

**Figure 2 molecules-29-05504-f002:**
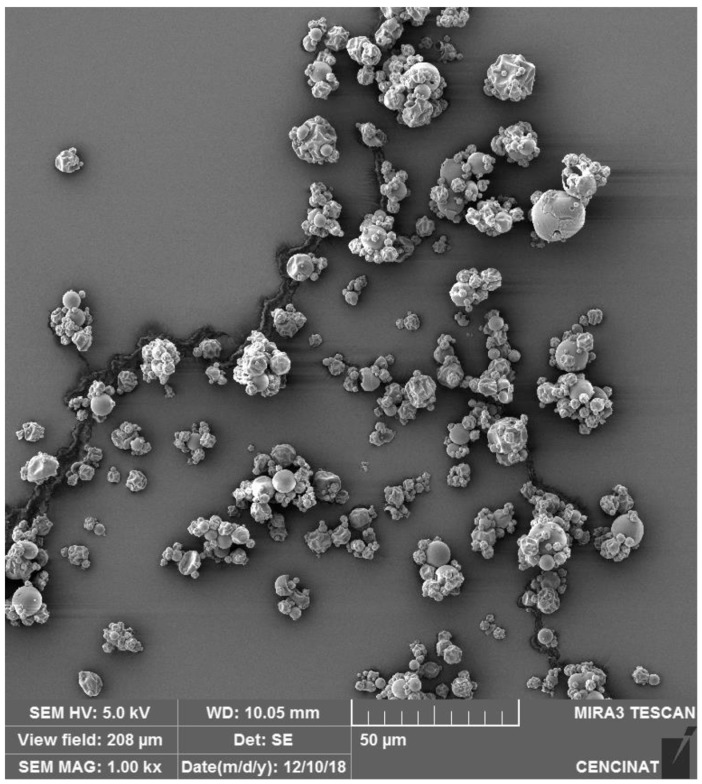
SEM image of *V. floribundum* microencapsulated particles prepared by spray-drying, captured with a TESCAN MIRA 3 SEM at 1.00 k× magnification and a 50 μm scale bar.

**Figure 3 molecules-29-05504-f003:**
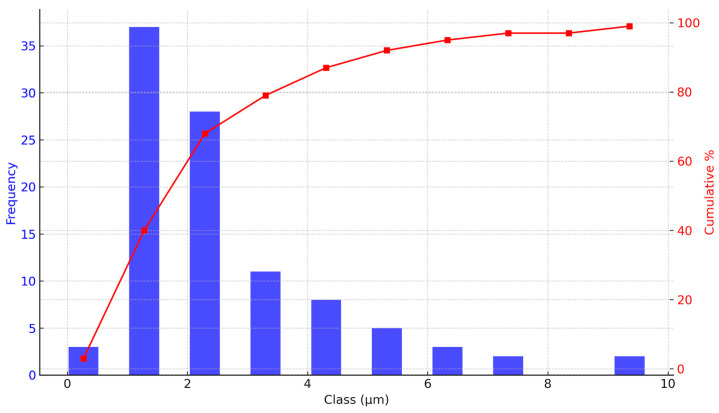
Particle size distribution analysis of *V. floribundum* microencapsulated spheres. Particle size distribution and cumulative distribution curve of *V. floribundum* microencapsulated spheres, measured using FIJI software (version 2.9.0). The histogram exhibits a positively skewed distribution with a peak at 1.28 μm. The cumulative curve shows that about 79% of the particles are smaller than 3.30 μm, indicating a concentration of small to medium-sized particles in the sample.

**Figure 4 molecules-29-05504-f004:**
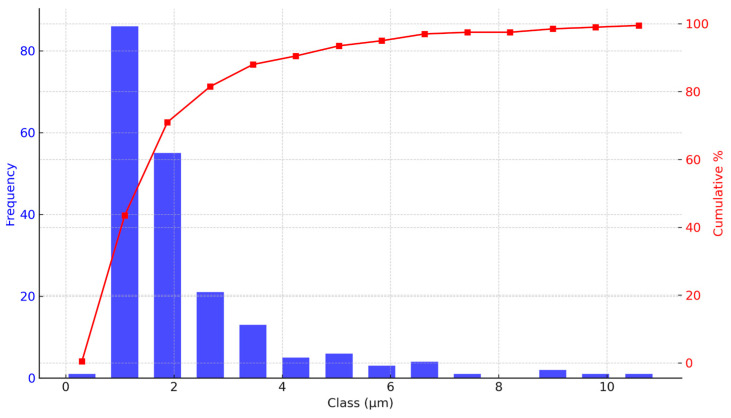
Particle size distribution analysis of *R. glaucus* microencapsulated spheres. Particle size distribution and cumulative distribution curve of *R. glaucus* microencapsulated spheres, analyzed using FIJI software (version 2.9.0). The histogram shows a positively skewed distribution with a mode of 1.09 μm. The cumulative curve indicates that approximately 80% of the particles are smaller than 3 μm, highlighting a predominance of fine particles in the sample.

**Figure 5 molecules-29-05504-f005:**
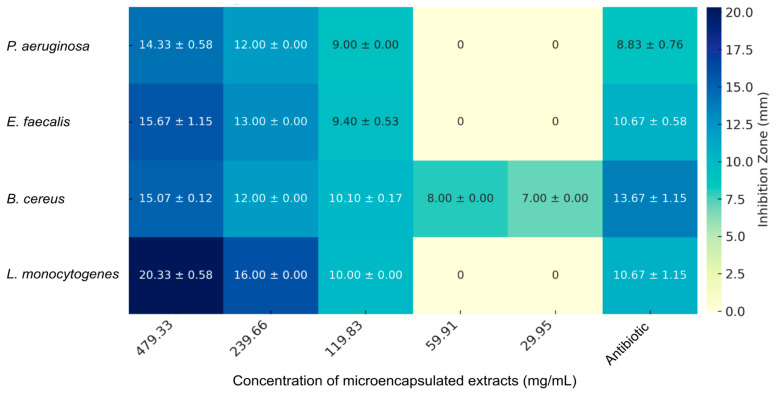
Antimicrobial activity of microencapsulated *R. glaucus* anthocyanin extract against different bacterial strains. Heatmap showing the inhibition zones (in mm) with standard deviations for *P. aeruginosa*, *E. faecalis*, *B. cereus*, and *L. monocytogenes* at various concentrations (479.33–29.95 mg/mL) of the microencapsulated *R. glaucus* extract. The heatmap illustrates a concentration-dependent inhibition, with notable effectiveness against *L. monocytogenes* and *B. cereus* at higher concentrations. Antibiotics were used at the standard concentration.

**Figure 6 molecules-29-05504-f006:**
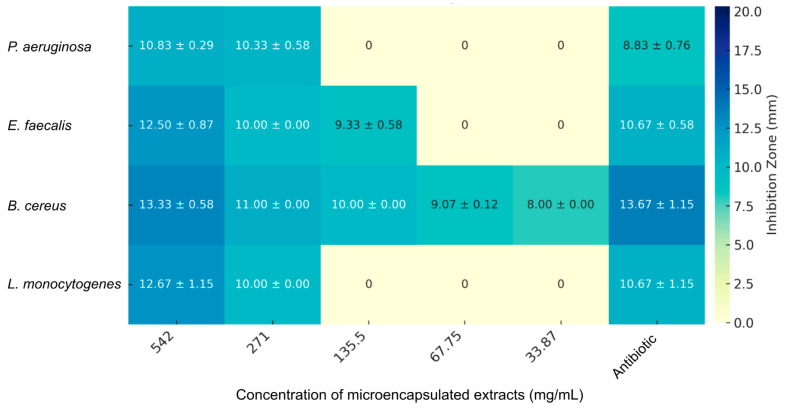
Antimicrobial activity of microencapsulated *V. floribundum* anthocyanin extract against various bacterial strains. The heatmap illustrates the inhibition zones (in mm) with standard deviations for *P. aeruginosa*, *E. faecalis*, *B. cereus*, and *L. monocytogenes* at different concentrations (ranging from 542 to 33.87 mg/mL) of the microencapsulated *V. floribundum* extract. The data reveal a concentration-dependent inhibitory effect, with significant antimicrobial activity observed mainly at higher concentrations, particularly against *B. cereus*. Antibiotics were used at the standard concentration.

**Figure 7 molecules-29-05504-f007:**
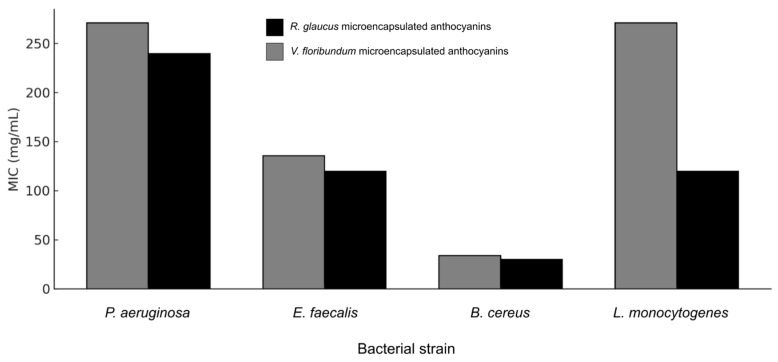
MIC of microencapsulated *R. glaucus* and *V. floribundum* extracts against various bacterial strains. The clustered bar chart shows the MIC values (in mg/mL) for each extract against four bacterial strains: *Pseudomonas aeruginosa*, *Enterococcus faecalis*, *Bacillus cereus*, and *Listeria monocytogenes*.

**Figure 8 molecules-29-05504-f008:**
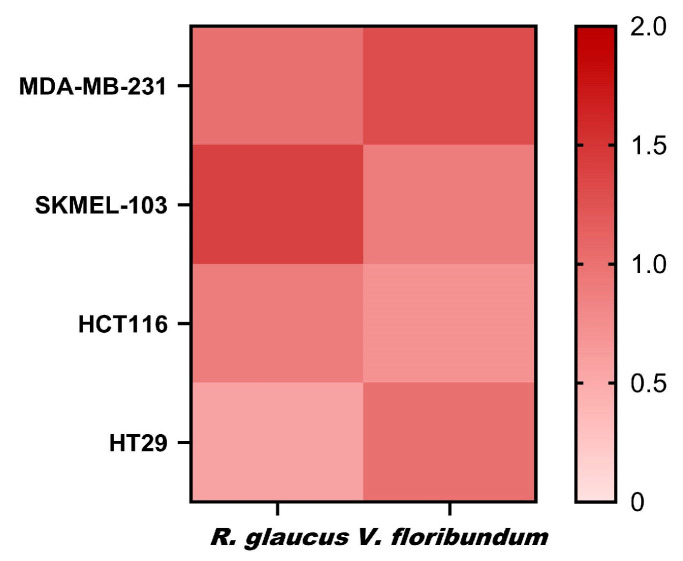
Therapeutic index (TI) values calculated as the ratio of IC_50_ of non-tumor cells to the IC_50_ in tumor cells.

**Figure 9 molecules-29-05504-f009:**
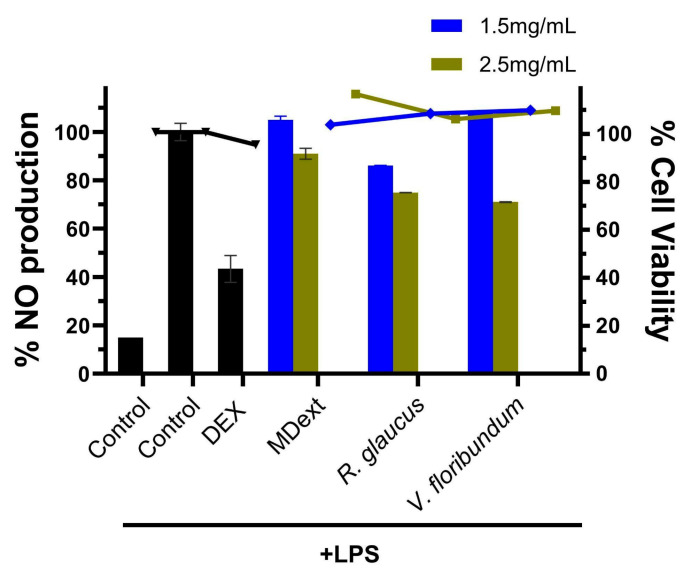
Anti-inflammatory activity of microencapsulated *R. glaucus* and *V. floribundum* extracts. RAW264.7 cells were pretreated with the sample compounds and then stimulated with LPS. The percentage of NO production was calculated using cells treated with LPS only (control + LPS) as 100% NO production (right *Y* axis). Dexamethasone (DEX) was used as positive control, and cells were incubated with only cell media as negative controls (control). Maltodextrin (MDext) was also included as a control. Bars represent the percentage of NO production (left *Y* axis). Dots and lines represent the % cell viability for each corresponding data set (right *Y* axis).

**Table 1 molecules-29-05504-t001:** Total polyphenols content (TPC) (gallic acid mg/g fresh weight of fruit), anthocyanins concentrations expressed as cyanidin 3-glucoside, and antioxidant activity (IC_50_ values corresponding to the DPPH assay expressed in μg/mL anthocyanins) of *V. floribundum* and *R. glaucus.* * Significant differences between *V. floribundum* and *R. glaucus* for TPC, anthocyanin content, and antioxidant activity (*p* < 0.001 in all cases).

Parameter	*V. floribundum*	*R. glaucus*
Total polyphenols content (TPC) (gallic acid mg/100 g fresh weight)	354 ± 25.16 *	294 ± 24.03 *
Anthocyanin content (mg/100 g)	79.67 ± 5.61 *	53.3± 3.9 *
Antioxidant activity of fruit extract (IC50 DPPH assay expressed in μg/mL anthocyanins)	83.5 ± 19.40 *	167.92 ± 39.57 *

**Table 2 molecules-29-05504-t002:** Summary of FTIR comparative visual analysis of *Vaccinium floribundum*. Comparative visual analysis of FTIR spectra of non-microencapsulated and microencapsulated anthocyanins from *Vaccinium floribundum*, and list of key wavenumbers corresponding to functional groups identified in the spectra, along with the qualitative intensity observed in non-microencapsulated and microencapsulated samples.

Wavenumber (cm^−1^)	Functional Group	Non-Microencapsulated (Qualitative Intensity)	Microencapsulated (Qualitative Intensity)
3200–3600	O-H stretching (hydroxyl)	High	Medium
2800–3000	C-H stretching (alkanes)	-	Medium
1700–1750	C=O stretching (carbonyl)	High	Low
1500–1600	C=C stretching (aromatic rings)	Medium	Low
1000–1300	C-O, C-O-C stretching (ethers, glycosidic bonds)	Medium	Low
1027	C-O-C stretching (polysaccharides)	-	Medium

**Table 3 molecules-29-05504-t003:** Summary of FTIR comparative visual analysis of *Rubus glaucus.* Comparative visual analysis of FTIR spectra of non-microencapsulated and microencapsulated anthocyanins from *Rubus glaucus* and wavenumbers associated with major functional groups and the qualitative intensity of these groups in both non-microencapsulated and microencapsulated samples.

Wavenumber (cm^−1^)	Functional Group	Non-Microencapsulated (Qualitative Intensity)	Microencapsulated (Qualitative Intensity)
3200–3600	O-H stretching (hydroxyl)	High	Medium
2800–3000	C-H stretching (alkanes)	Low	Medium
1700–1750	C=O stretching (carbonyl)	Medium	Low
1500–1600	C=C stretching (aromatic rings)	Medium	Low
1000–1300	C-O, C-O-C stretching (ethers, glycosidic bonds)	Medium	Low to Medium
1027	C-O-C stretching (polysaccharides)	-	Medium

**Table 4 molecules-29-05504-t004:** Inhibitory concentration values (IC_50_) (mg/mL) of microencapsulated extracts against tumor and non-tumor cell lines at 72 h. *** indicates significance differences between the microencapsulated extracts (*p* < 0.001).

	MDAMB231	SKMEL103	HCT116	HT29	NIH3T3
*R. glaucus*	4.16 ± 0.25 ***	3.07 ± 0.60 ***	5.03 ± 0.09 ***	4.76 ± 0.20 ***	4.32 ± 0.37 ***
*V. floribundum*	8.15 ± 1.67	15.46 ± 1.01	15.59 ± 3.55	10.53 ± 0.22	10.44 ± 1.57

## Data Availability

Data are contained within the article and Appendix A.

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
