# Peer review of "Bioactive Properties of Microencapsulated Anthocyanins from Vaccinium floribundum and Rubus glaucus"

_molecules, 2024, doi:10.3390/molecules29235504_

Round 1
Reviewer 1 Report
Comments and Suggestions for Authors
- The study is a comprehensive investigation into the morphological and antibacterial properties of microencapsulated anthocyanin extracts from Rubus glaucus and V. floribundum prepared using spray drying. It effectively combines morphological characterization, particle size analysis, and evaluation of antibacterial activity, providing valuable insights into the efficacy and potential applications of the encapsulated anthocyanins. Below, I summarize some recommendations regarding its acceptance. With these revisions, the study will make a meaningful contribution to the fields of food technology and pharmaceutical applications, specifically in natural product encapsulation and bioactivity evaluation. The innovative application of SEM and antibacterial testing, alongside practical insights for optimization, makes this work a valuable resource for researchers and industry professionals alike.
- ABSTRACT
-
Line 20-21: "Highly susceptible to degradation due to environmental factors such as light, temperature, and pH." Consider briefly mentioning why this degradation matters (e.g., "leading to reduced bioavailability and efficacy").
-
Line 23-24: "Microencapsulation, a protective technique involving the entrapment of anthocyanins in a protective matrix..." The phrase "protective technique" and "protective matrix" is somewhat redundant. Rephrasing to "Microencapsulation, which involves entrapment in a matrix to enhance stability and bioavailability..." would be more concise.
-
Line 26-27: "Using maltodextrin as the carrier agent through a spray-drying process." Consider specifying why maltodextrin was chosen—e.g., "due to its film-forming properties and effectiveness in stabilizing sensitive compounds."
-
Line 29-30: "The biological activities of these microencapsulated anthocyanins were evaluated for their antibacterial, antioxidant, and anti-inflammatory effects." It might be helpful to add a brief mention of how these activities were measured, for instance: "using in vitro assays."
-
Line 30-31: "The results indicated enhanced bioactivity of the microencapsulated anthocyanins." This is a key point that could benefit from more specificity. For example, "The results indicated enhanced antioxidant, antibacterial, and anti-inflammatory activity of the microencapsulated anthocyanins compared to the non-encapsulated form."
-
Line 33-34: "This study provides valuable insights into the effectiveness of microencapsulation in preserving anthocyanins' functional properties and enhancing their health-promoting effects." To make this more impactful, consider adding, "highlighting the potential for application in the food and pharmaceutical industries."
- INTRODUCTION
-
Lines 41-42: Consider including a specific example of well-known fruits or vegetables to illustrate the broad presence of anthocyanins and make it more engaging for readers.
-
Line 44: Change "have been associated to the prevention" to "have been associated with the prevention." This will improve the precision of the language.
-
Lines 49-50: It is recommended to briefly explain how microencapsulation "controls release in the body" to provide a clearer idea of the mechanism to non-specialized readers.
-
Line 61: Change "in the development in functional foods and pharmaceutical development" to "in the development of functional foods and pharmaceutical products." This correction will help avoid redundancy and improve the clarity of the message.
-
Expand on the State of the Art:
Consider including a more detailed review of recent studies on anthocyanin microencapsulation, focusing on the specific challenges that the present study aims to address. -
Clarify the Study's Innovation:
Explain in more detail what makes this study innovative compared to existing methods. For example, is there any specific characteristic of the microencapsulation with maltodextrin that offers significant advantages? Or why was the spray-drying process chosen over other encapsulation methods? - RESULTS AND DISCUSSION
- Comparative Analysis: While the individual analysis of both V. floribundum and R. glaucus is well-executed, a more direct comparison between the two species could improve the reader's understanding of the differences. Explicitly highlighting why the particle size distribution of R. glaucus is slightly narrower than V. floribundum might offer additional insight into the influence of different fruit properties on encapsulation.
- Optimization Recommendations: The authors mention that factors like feed concentration, drying temperature, and atomization rate can be optimized, but the discussion would benefit from specific examples or values that have been found effective in similar studies. Providing more targeted recommendations could enhance the practical utility of the findings for future research or industrial application.
- Link to Bioactive Properties: The relationship between particle morphology and bioactive properties (e.g., stability or release rate of anthocyanins) could be more thoroughly discussed. For instance, explaining how the observed particle morphology influences the antioxidant or antibacterial efficacy would provide a more comprehensive understanding of the practical benefits of these findings.
- Comparison with Non-Microencapsulated Extracts: It would be beneficial to include a comparison of the antibacterial activity of non-microencapsulated anthocyanins. This would help to determine whether microencapsulation has enhanced or retained the antibacterial properties of the extracts.
- Mechanistic Insights: The section would be strengthened by discussing potential mechanisms behind the observed antibacterial activity. For example, are the anthocyanins likely disrupting bacterial cell membranes, or do they interfere with intracellular components? Citing relevant literature here could provide more depth to the discussion.
- Statistical Analysis: While standard deviations are provided, the use of statistical tests (e.g., ANOVA or t-tests) to determine the significance of differences between treatment groups would strengthen the conclusions. This would provide more confidence in asserting whether differences in inhibition zones are statistically meaningful.
- Broader Implications: Expanding on the broader implications of these antibacterial findings would be valuable. For instance, could these microencapsulated anthocyanins serve as natural preservatives in food products, or as potential therapeutic agents against antibiotic-resistant bacteria? Including such considerations would help contextualize the potential real-world applications of the research.
Author Response
Abstract:
- We have revised the phrases regarding anthocyanin degradation, microencapsulation, and the biological activities of the microencapsulated anthocyanins to provide more clarity. Specifically, we have added explanations for why degradation affects bioavailability and efficacy, adjusted redundant phrases, and mentioned the use of in vitro assays to evaluate biological activities.
- Additionally, we clarified the enhancement of bioactivity in the microencapsulated form compared to non-encapsulated anthocyanins, and we emphasized the potential applications in the food and pharmaceutical industries. Please find all the changes highlighted in yellow.
Introduction:
- We expanded the introduction by adding examples of fruits and vegetables rich in anthocyanins and by explaining the role of microencapsulation in protecting anthocyanins from degradation while also controlling their release in the body.
- We have also clarified the innovation of the study, particularly the use of maltodextrin as the encapsulating matrix, and discussed the advantages of the spray-drying process in producing uniform microspheres.
- Please find all the changes highlighted in yellow.
Results and Discussion:
- We added a comparative analysis to explain the narrower particle size distribution of R. glaucus compared to V. floribundum, suggesting that differences in anthocyanin composition and extract viscosity influenced droplet formation during spray drying.
- In response to your suggestion on optimization, we included a discussion on how adjusting feed concentration and spray-drying temperatures, based on similar studies, could improve encapsulation efficiency and particle size uniformity.
- We also provided more detailed mechanistic insights, including the contribution of particle morphology to the stability and bioactivity of the microencapsulated anthocyanins. Specifically, we discussed the membrane-disruptive effects of anthocyanins as a potential explanation for the enhanced antibacterial activity observed.
- We added a new section in Materials and Methods to include details on statistics.
- Finally, we expanded the discussion on the broader implications of these findings, suggesting the potential of microencapsulated anthocyanins as natural preservatives and therapeutic agents against antibiotic-resistant bacteria.
- Please find all the changes highlighted in yellow.
Reviewer 2 Report
Comments and Suggestions for Authors
Overall, the manuscript is well-written and organized. My biggest concern concerns the controls used, which need to be clarified. Please review your M&M section. For additional comments and suggestions, please take a look at the file I've attached.

Author Response
- Is there any published studies on the anthocyanin content of these two native fruit from Ecuador?
Thank you for your comment. Yes, there are published studies on the anthocyanin content of these two native fruits from Ecuador. We have cited our previous work, which evaluates the biological activity and anthocyanin content of R. glaucus and V. floribundum in their non-microencapsulated form. We included this reference in the revised manuscript to provide the relevant background and ensure proper citation of the existing data on these fruits
- Besides, the fruit you used in the study were not freshly harvested. If they were bought at the market, you don't know how conditions (mostly time-temperature) were between harvest and display at the store. Supply chain conditions may also contribute to anthocyanin degradation.
We acknowledge the reviewer’s point regarding the potential degradation of anthocyanins due to unknown supply chain conditions between harvest and market display. This is indeed a valid concern and could contribute to the lower polyphenol and anthocyanin content observed in our study. We did not have control over the time-temperature conditions before the fruits were purchased, and these factors may have impacted the integrity of the bioactive compounds. We have now included this limitation in the discussion section, recognizing that future studies using freshly harvested fruits directly from the source would allow for a more accurate assessment of their bioactive properties and avoid any potential degradation caused by supply chain factors.
- If the fruit were not harvested and handled under the same exact conditions, then differences in fruit chemical composition are expected. So, when comparing results one needs to be careful about the origin and postharvest conditions before analysis. For accuracy regarding fruit composition, the fruit should have been freshly harvested and composition determined shortly after.
We appreciate the reviewer’s observation regarding the importance of consistent harvest and postharvest conditions when comparing results. While the fruits in our current and previous studies were not harvested under identical conditions, we acknowledge that variations in origin and postharvest handling may have contributed to differences in chemical composition, including antioxidant activity. We have clarified this point in the discussion, noting that future studies could benefit from using freshly harvested fruits under controlled conditions to ensure a more accurate comparison of bioactive compound content and activity.
- Statistically significant? Please see comment om M&M section; statistical analysis method description missing.
We appreciate the reviewer’s comment regarding the need for clarification on the statistical significance of the values presented in Table 1. In the modified version of the manuscript, we included a detailed description of the statistical tests used to determine the significance of the differences between the values, as well as the corresponding p-values. We also indicate with an asterisk in the tables which differences are statistically significant. Finally, we included a phrase describing the results from the statistical analysis.
- Can you discuss further the mechanisms behind anthocyanin degradation and encapsulation? Is the matrix acting as a coating or is there any interactions or synergistic effect between the matrix and anthocyanins?
We appreciate the reviewer’s insightful question regarding the mechanisms behind anthocyanin degradation and the role of the encapsulating matrix. Anthocyanins are highly susceptible to degradation due to environmental factors such as pH, light, and temperature, primarily because of their reactive hydroxyl groups. In our study, the encapsulating matrix (maltodextrin) likely acts as a protective coating, reducing the exposure of these reactive groups and preventing their interaction with environmental stressors. Additionally, the reduction or absence of the O-H stretching bands in the FTIR spectra suggests a potential interaction between the matrix and anthocyanins, such as hydrogen bonding, which could further stabilize the anthocyanins by limiting their reactivity. This encapsulation mechanism not only protects anthocyanins from degradation but also enhances their stability and bioavailability. We included this discussion in the manuscript to further clarify the potential synergistic effects between the matrix and the anthocyanins.
- How does maltodextrin protects anthocyanins from degradation?
We have added an explanation of how maltodextrin protects anthocyanins from degradation in the newly included paragraph in the discussion section, addressing the previous suggestion.
- What are the changes in anthocyanin structure due to maltodextrin incorporation?
We have addressed the changes in anthocyanin structure due to maltodextrin incorporation in the revised discussion section. In summary, the incorporation of maltodextrin primarily results in a reduction of O-H stretching bands in the anthocyanins, indicating decreased exposure of reactive hydroxyl groups. This suggests hydrogen bonding between maltodextrin and anthocyanins, which stabilizes the anthocyanins without causing major chemical modifications to their core structure.
- How does the differences in size and shape of particles affects the efficacy of microencapsulation?
We appreciate the reviewer’s comment regarding the effect of particle size and shape on the efficacy of microencapsulation. Variations in particle size and shape can significantly influence the encapsulation efficiency, stability, and controlled release properties of the encapsulated compounds. Smaller and more uniform particles generally provide a larger surface area, which can improve dissolution rates and bioavailability. In contrast, larger or irregularly shaped particles may lead to reduced encapsulation efficiency and slower release of the active compounds. In our study, the observed differences in particle morphology between the two berry species may reflect slight variations in encapsulation performance, which we acknowledge as a factor to be optimized in future research, we have included a paragraph in the discussion for more clarification.
- How would you adjust these parameters? That is, what would be the best spray-dry conditions compared to those you chose for this study?
We appreciate the reviewer’s comment regarding the optimization of spray-drying parameters. Based on similar studies, we suggest that increasing the feed concentration to 20-25% solid content and maintaining the drying inlet temperature between 140-160°C could potentially improve encapsulation efficiency and particle size uniformity. Additionally, adjusting the atomization rate to achieve smaller droplets could help produce more uniform particles. These parameters could be optimized in future experiments to enhance encapsulation quality and consistency. We have added a paragraph to further clarify this in the discussion.
- Why are the particles so different between R. glaucus and V. floribundum?
The differences in particle morphology between R. glaucus and V. floribundum may be attributed to variations in the chemical composition of the anthocyanins and the viscosity of the extracts from each species. These factors influence droplet formation during spray drying, resulting in different particle sizes and shapes. Future studies could explore how adjusting spray-drying parameters for each specific extract may reduce these differences. We have added a paragraph to further clarify this in the discussion.
- The same encapsulation procedure seemed to have been used for both fruit species (from M&M). So, please further explain why the differences in particle size between the two species.
We appreciate the reviewer’s follow-up on this point. As mentioned in a previous response, although the same encapsulation procedure was applied to both fruit species, the differences in particle size are likely due to variations in the chemical composition and viscosity of the anthocyanin extracts from each species. These factors can influence droplet formation during the spray-drying process, leading to differences in particle morphology. Future studies could explore adjustments in spray-drying parameters, such as feed concentration and atomization rate, to minimize these variations.
- Maltodextrine alone, without the extract?
Yes, maltodextrin was tested alone, without the extract, at the maximum concentration of 550 mg/mL to demonstrate that it does not exhibit any antibacterial activity on its own. This was done to confirm that the observed growth inhibition in the microencapsulated samples was solely due to the bioactive components from the extract and not from the maltodextrin itself.
- Do you have a control? No extracts?
Yes, we included a control using maltodextrin alone, without the extract, at the maximum concentration tested (550 mg/mL). This control confirmed that the maltodextrin itself does not exhibit any antibacterial activity, ensuring that the observed inhibition zones in the microencapsulated samples are solely due to the bioactive components from the extract.
- Please add a legend into the graph, so you do not have to show this information in the figure caption.
Thank you for the suggestion. We added a legend directly into the graph in Figure 7 to provide the necessary information, which will allow us to streamline the figure caption.
- Statistically significant? Please show statistics.
We have performed statistical analyses to determine significant differences between the microencapsulated extracts. The results are now included in Table 4 and Figure S3.
- Control against MDdext + extracts?
Thank you for the question. Yes, we included maltodextrin (MDext) alone as a control. This allowed us to confirm that the observed effects on NO production and cell viability were due to the bioactive compounds in the extracts and not maltodextrin itself. The results showed that maltodextrin alone did not significantly affect NO production or cell viability.
- Material and Methods. Description of statistic analysis methodology used is missing.
Thank you for pointing this out. In the modified version of the manuscript, we included the section 3.12 Statistical analysis, detailing the methods used, specifying the tests applied and the criteria for determining statistical significance (e.g., p-value thresholds).
- How many fruit were bought and how many fruit from each species were selected for the study? How many biological replicates?
Thank you for your question. In our study, we purchased 5kg of each fruit species (R. glaucus and V. floribundum). From these, 3kg of fruits from each species were selected based on size, ripeness, and absence of visible damage. We performed 3 biological replicates for each assay to ensure reproducibility of the results. We added this information in the Materials and Methods section
- How? Details about the procedure used.
Thank you for your observation. We have added the details in the Plant material section regarding the freeze-drying and pulverization process in the Methods section. This includes the specific conditions for freezing, drying, and grinding the fruit samples to ensure clarity and reproducibility of the procedure.
- Details
Thank you for your question. Anthocyanins were extracted using a Rotine 380 centrifuge (Hettich, Germany) at 6,000 RPM for 15 minutes at 4°C to ensure efficient separation and prevent degradation of the bioactive compounds. This information was added to the manuscript.
- How were the fruit lyophilized? details about instrumentation.
Thank you for your question. The fruits were stored at -20°C and freeze-dried using a Labconco FreeZone® 6 Liter Benchtop Freeze Dryer. The text was added to the Plant material section.
- Detail about instrument used to measure absorbance.
Thank you for your observation. We have now included the details of the spectrophotometer used to measure absorbance in the Methods section. Specifically, a Thermo Scientific Genesys 10S UV-Vis spectrophotometer. was used for the measurements at 515 nm and 700 nm. The information was added to section 3.3
- Were non-microencapsulated fruit used as control? How were they prepared? Did you have a treatment with carrier agent only?
- Control (non-encapsulated): Thank you for your insightful suggestion. The primary objective of this study was to assess the potential benefits of microencapsulation on the stability and biological activity of anthocyanins from R. glaucus and V. floribundum. For this reason, we focused on comparing the microencapsulated anthocyanin extracts with the carrier agent (maltodextrin) as a control to demonstrate that the observed effects were due to the anthocyanins and not the carrier itself.
Additionally, we have previously published a comprehensive study on the biological activity of non-microencapsulated extracts from these fruits, which provides detailed data on their efficacy in their native form. Thus, including non-microencapsulated fruit extracts as a control was not necessary in the context of this study, as the primary goal was to evaluate the impact of the microencapsulation process itself.
We believe that the current experimental design, focusing on the comparison between the microencapsulated anthocyanins and the carrier agent control, sufficiently addresses the objectives of this study. We will, however, include a reference to our previous publication in the discussion to guide readers seeking information on the biological activity of non-microencapsulated extracts.
- Encapsulated (carrier agent + anthocyanins). Yes it is included.
- Encapsulated (carrier only): Thank you for your observation. A control with maltodextrin only (carrier only) was not included in this study because the primary objective was not to standardize microencapsulation parameters or to explore the use of maltodextrin as a carrier. These aspects of microencapsulation, including the use and conditions of maltodextrin, have been well-documented in previous studies.
- How many fruit per treatment were used? How many replicates per treatment? Treatments and controls need to be better explained.
Microencapsualtion was performed with 600g of liophylized fruit, then were used for the different treatments and controls. Each treatment was performed with three biological replicates to ensure reproducibility. We will update the Methods section to further clarify it.
- How was the pigment concentrate prepared?
Thank you for your observation. The term 'pigment concentrate' refers to the anthocyanin extract obtained from the lyophilized fruit. The preparation of the pigment concentrate involved first freeze-drying the fruit, followed by extraction under the conditions described in Section 3.2 of the Methods. The extract was then concentrated by rotary evaporation. To improve clarity, we have replaced the term 'pigment concentrate' with 'anthocyanin extract' in the manuscript. This term more accurately reflects the anthocyanin-rich extract obtained from the lyophilized fruit, as described in the Methods section.
- Why were these conditions chosen?
Reading the Results section it seems that these conditions need to be futher optimized for better encapsulation uniformity. Maybe using different spray-dry conditions and compare the results.
Thank you for your comment. The spray-drying conditions (inlet temperature of 150°C and outlet temperature of 90°C) were chosen based on previous studies that demonstrated these parameters as effective for preserving the stability of anthocyanins while achieving efficient drying. These conditions were selected to balance the preservation of bioactive compounds and the production of a stable microencapsulated powder. The reference was added in the manuscript.
However, we acknowledge that the results indicate some variability in encapsulation uniformity, suggesting that further optimization may be beneficial. Future studies could explore varying spray-drying conditions, such as adjusting feed concentration, atomization rate, or temperature, to enhance encapsulation efficiency and particle uniformity. We included this point in the discussion as a potential area for improvement in future work.
- Why were different amounts used?
The different amounts used (479 mg/mL for R. glaucus and 542 mg/mL for V. floribundum) correspond to the maximum solubility levels of the microencapsulated samples. We have added this information to clarify the methodology section.
- How? Details on equipment used to maintain such conditions. Instrument details regarding absorbance measurement.
We have included additional details regarding the equipment used to maintain cell culture conditions and the instrument specifications for measuring absorbance in the Materials and Methods section.
Round 2
Reviewer 1 Report
Comments and Suggestions for Authors
Thank you for submitting the revised version of your manuscript. After reviewing the corrections you've made, I am pleased to inform you that I believe the revisions adequately address the feedback provided during the review process.
As a result, I will be recommending acceptance of your article for publication. Please note, however, that the final decision will depend on the evaluations from the other reviewers as well.
You will be notified once the full review process is complete. In the meantime, thank you again for your careful revisions and for your contribution to our journal.
Author Response
Thank you very much for your positive feedback and for recommending the acceptance of our manuscript. We are grateful for your valuable suggestions and guidance throughout the review process, which have undoubtedly improved the quality of our work.
We look forward to hearing the final decision once the full review process is complete. Thank you once again for your time and consideration.
Best regards